# Dimensions, Measures, and Contexts in Psychological Investigations of Curiosity: A Scoping Review

**DOI:** 10.3390/bs12120493

**Published:** 2022-12-02

**Authors:** Yong Jie Yow, Jonathan E. Ramsay, Patrick K. F. Lin, Nigel V. Marsh

**Affiliations:** School of Social and Health Sciences, James Cook University, Singapore 387380, Singapore

**Keywords:** curiosity, dimensions of curiosity, measures of curiosity, contexts of curiosity

## Abstract

The study of curiosity as a construct has led to many conceptualisations, comprising of different dimensions. Due to this, various scales of curiosity have also been developed. Moreover, some researchers have conceived of curiosity as a general trait-like, while others have included contexts, such as the workplace, or education when investigating curiosity. This scoping review aims to scope the extant psychological literature on curiosity in order to better understand how it has been studied, specifically with regard to its dimensions, measures, and contexts. A total of 1194 records were identified, with 245 articles meeting the inclusion criteria. Results suggest that the majority of curiosity research examined curiosity as having multiple dimensions and analysed the dimensions individually, with a deprivation-type curiosity playing the biggest role. The measure most commonly used was the Epistemic Curiosity Scale, which also consisted of a deprivation-type curiosity as one of the dimensions. Findings also implied that curiosity was most studied in the context of the workplace. Supplementary findings included a lack of representation of non-Western countries, as well as needing to cross-validate a recently developed curiosity scale. This scoping review represents a consolidation of the curiosity literature and how it can further prosper.

## 1. Introduction

The benefits of curiosity cannot be understated. Prior work has shown that curiosity is linked to multiple positive outcomes, such as happiness, meaning in life, distress tolerance, and satisfying social relationships [1]. At the workplace, curiosity has also been positively associated with job performance and a sense of belonging to organizations [2,3]. In education, curiosity aids learning and subsequently improves academic performance [4]. While possessing curiosity is undoubtably beneficial, research into the construct of curiosity is fragmented. For example, there are many conceptualizations of curiosity based on its structure, such as having two dimensions, or five dimensions [5,6]. Different contexts (e.g., workplace, education, or no specific context) have also been examined when investigating curiosity, raising questions as to whether there is added validity in using context-specific measures. As a result, depending on how curiosity was conceptualised, a variety of measures have been developed and used to capture this seemingly multifaceted construct. As such, the purpose of this scoping review is to comprehensively scope the extant psychological literature on curiosity in order to better understand how it has been studied, specifically with regard to its dimensions, measures, and contexts. The scoping review also seeks to identify productive areas for future research.

### 1.1. Dimensions and Measures of Curiosity

Over the years, researchers have sought to unravel the construct of curiosity. Early conceptualizations developed by philosophers such as Aristotle and Kant positioned curiosity as an intrinsically motivated desire for new information, or an appetite for knowledge [7,8]. While these conceptualizations are still applicable in today’s context, curiosity has since been seen as a component of personality by contemporary researchers in psychology. For example, personality researchers have placed curiosity as part of the web of personality facets under personality inventories like the NEO-PI-R, where curiosity is an element within the Openness trait, and the fantasy and aesthetic sub-facets [9]. This is similar to the Aesthetic Appreciation facet in the HEXACO personality inventory [10]. However, the HEXACO moved beyond the aesthetic conceptualisation and consisted of an inquisitiveness facet, which is defined as the tendency to seek information and experiences with the natural and human world, where high scorers are curious about politics, and low scorers have little interest in reading encyclopaedias, confirming that curiosity has more to offer than that of an artistic nature. 

Moving beyond conceptualisations of curiosity as a component of broader personality traits, other researchers have examined curiosity more directly, some seeking to characterize it as a singular trait while others seek to delineate various forms of curiosity. In this vein, some researchers have examined curiosity as a unidimensional trait. This could be due to researchers adopting the view that curiosity is simply an overall interest or inclination to find out about something in general, such as in Galli et al., where the researchers examined participants’ trait-level curiosity [11]. Other researchers also used ad hoc created scales such as single item measures to capture curiosity in a general and overall (i.e., single-dimension) manner [12].

In contrast to the unidimensional approach, other researchers have conceptualised curiosity as a multidimensional construct with distinct components [6], seeking to identify different forms of curiosity that may to some extent vary independently. However, the literature is not aligned on how the dimensions of curiosity should be defined and described. For example, a dimension that commonly appears is an intrinsic motivation to seek knowledge, where gaining knowledge yields positive emotions such as joy. Kashdan et al. describe this dimension as a general fascination with new information that leads to positive emotions, and term it stretching (part of the Curiosity and Exploration Inventory; CE-II) [1]. More recently, Kashdan et al. referred to this dimension as joyous exploration, and named it as one of the five dimensions of curiosity in their Five-Dimensional Curiosity Scale (5DC) [6]. Litman and Spielberger provided a similar definition to this dimension, and called it diversive epistemic curiosity, which was derived from the Epistemic Curiosity Inventory [5]. Evidently, this facet often manifests as different names, yet seemingly referring to the same dimension. While the nomenclature differs, it appears that this facet points to an approach-type curiosity, or more specifically, an interest-type curiosity. Individuals exhibiting this interest type curiosity would seek knowledge out of their own volition, and subsequently gain pleasure from learning new information.

Other than an interest-type curiosity, a deprivation-type curiosity was proposed by Loewenstein, who purported that an individual could be curious, not due to an interest in acquiring new knowledge, but rather more from a desire to learn what one does not know, or to close the information gap [13]. Loewenstein’s conceptualisation suggests that information gaps cause anxiety and pique curiosity in a bid to resolve the gap in knowledge. Many studies examined this deprivation-type curiosity [14,15]. Independently developed scales have also been used to measure this deprivation facet (e.g., the deprivation subscale of the Epistemic Curiosity Scale; the deprivation-sensitivity subscale of the 5DC) [6,16]. In contrast with interest-type and approach-oriented curiosity, deprivation type curiosity is consistent with curiosity of an avoidant nature, where curiosity is displayed because individuals seek to lower anxiety from a lack of knowledge.

Moving beyond interest and deprivation type curiosity, researchers have also reported other conceptualisations, such as interpersonal curiosity, where individuals desire information about other people [17]. This is not unlike the social curiosity facet as part of the 5DC [6]. Again, this phenomenon appears to occur in other dimensions of curiosity, such as self-curiosity being equivalent to intrapersonal curiosity, where individuals are inquisitive about themselves [18,19]. This raises a potential issue: it is difficult to know how best to measure and investigate curiosity given the variety of conceptualizations and accompanying measures that are available. Over 30 measures exist in the literature, some thoroughly validated, and others created for usage in an isolated study [20]. As a result, it can be difficult to select a ‘best’ measure to administer when seeking to examine the antecedents, correlates, and consequences of curiosity. These measures of curiosity vary in terms of their dimensionality, some being unidimensional while others are multidimensional, while researchers adopting a multidimensional measure also vary in terms of analytic strategy, with some examining specific sub-dimensions of curiosity while others are happy to aggregate to an overall score. Though the optimal measure to administer will vary depending on the research question, it would nevertheless be useful to comprehensively survey the various conceptualizations and measures of curiosity in the extant literature, with a view to enabling researchers to make more informed choices when planning their investigations of curiosity. 

### 1.2. Contexts of Curiosity

Moving away from dimensions and measures of curiosity, the context or setting can play a role in its elicitation and expression. Research has found that context specificity is important when assessing various psychological constructs, and also advocated the use of frames of reference to increase the reliability and validity of measurement instruments [21,22]. In the case of curiosity, a person who is curious about their colleague at the workplace may not be curious about their teacher at school. The same person may be curious about the discipline of geography but not mathematics. Consistent with this idea, it has been argued that curiosity is context-specific [23]. Although it has been found that individuals differ in the degree of context-general curiosity, it is somewhat limiting to only examine overall curiosity, as individuals carry both context-general and context-specific individual differences across multiple settings [24]. Investigating specific contexts of curiosity has been shown to bring about additional benefits. For example, empirical evidence from research utilizing a workplace-specific curiosity scale indicates that work curiosity predicts job performance above and beyond more general predictors such as personality traits, cognitive ability, and integrity [25]. Additionally, Park et al. described that curiosity in a sporting context is different from general curiosity, and contexts outside of sports, and made one of the first attempts to determine if a facility can induce curiosity [26]. They found that curiosity about a sport facility (e.g., “I enjoy exploring brand new sport stadiums or facilities”) was a facet of curiosity in their sports curiosity measure. Research has pointed out that compared to other industries, one critical and unique feature of the sporting industry is facilities such as stadiums [27]. Accordingly, measuring sports curiosity specifically can provide a more apt representation of sports fans, compared to using a general curiosity scale. Likewise, a curiosity measure situated in the context of nature and its potentially dangerous elements (e.g., volcanic eruptions, hurricanes, and snowstorms) was created and purported to help explain motivation and behaviour specifically when nature is a factor [28]. As a precursor to potentially deep diving into the applicability of context-general and context-specific curiosity, this scoping review aims to summarise the contexts that curiosity has been studied in.

### 1.3. Rationale and Research Questions

The current literature conceptualizes curiosity in many ways. It has been studied as a facet-level personality construct under the broad trait of openness to experience, while researchers who focus more specifically on curiosity have variously conceptualized it as being unidimensional or as multidimensional, with the latter conceptualization having many different possible structures and domains. When treating curiosity as a multidimensional construct, some researchers focus on specific dimensions of curiosity, while others ultimately ignore the inherent multidimensionality and used aggregated total scores in their research. Moreover, some conceptualisations are seemingly similar, but are termed differently (interest-type curiosity vs. joyous exploration). While some facets are clearly delineated from one another (e.g., joy of learning new information vs. anxiety from not knowing), the distinction is less clear in others, and could be referring to the same underlying dimension (e.g., social curiosity vs. interpersonal curiosity, intrapersonal curiosity vs. self-curiosity). The presence (or absence) of contexts is also telling of the many ways that curiosity has been studied. 

Given the number of ways that curiosity has been conceptualized and measured, it is important to survey the various forms of curiosity, providing a functional overview of their prevalence and the various ways in which they have been employed in psychological research. Such an effort can help researchers make better informed decisions regarding the operationalization of the construct. In recent years, reviews of the curiosity literature have tended to focus on specific contexts like human resource development in organizations and education [20,29]. The current scoping review intends to cast a wider net to obtain a broader overview of research conducted thus far. Specifically, we sought to answer the following questions:To what extent is curiosity seen as being unidimensional or multidimensional?If curiosity is seen as being multidimensional, what are the dimensions that make up curiosity?How is curiosity measured?Is curiosity characterised as being context-general or specific?If curiosity is seen as being more specific, in which contexts has curiosity typically been studied?

## 2. Method

### 2.1. Framework

The framework for this scoping review was developed according to the Preferred Reporting Items for Systematic Review and Meta-Analyses (PRISMA). The Joanna Briggs Institute (JBI) approach to scoping reviews was also used as an additional resource when developing this scoping review [30]. A scoping review protocol documented the objectives, inclusion criteria, and methods of extraction, and was preregistered via deposit in the Open Science Framework (https://osf.io/dj7xg/; accessed on 8 July 2021)

### 2.2. Inclusionary Criteria

The PCC framework (Population, Concept, and Context) was used to develop the inclusion criteria, as recommended by JBI. Firstly, participants of all types were included in the scoping review. Regarding the concepts to be included, studies must focus on the psychological construct of curiosity, and have explicitly measured and analysed it as a variable of interest. Studies that utilise full-scale curiosity measures were included. Studies that measured only specific dimensions of curiosity were included as well (e.g., social curiosity dimension of the 5DCR). In the curiosity literature, self-report questionnaires are the most common approach to measuring curiosity [29]. Thus, we chose to include only quantitative research, focusing on empirical papers using self-report surveys, while being open to all types of study designs (e.g., experimental, quasi-experimental, correlational). All contexts (e.g., workplace, education) and article types (e.g., journal articles, theses, conference papers) were qualified for inclusion, and no exclusion criteria was imposed on the timeframe of records. The multidisciplinary databases of Scopus, Web of Science, and the ProQuest platform were used to conduct the scoping review. Lastly, only records in the English language were included.

### 2.3. Search Strategy

To begin, an initial search was conducted on Scopus in June 2021 using the keywords: (Curiosity) AND (facet OR dimension). From the results, relevant articles were picked up and analysed for further keywords to refine the search. A keyword analysis tool was also used to aid the process [31]. Following an iterative process, we decided that the word ‘curiosity’ had to be specified in the article title in order to yield articles where curiosity was a specific, measured variable of interest. As second level keywords, we searched for the terms ‘facet’ OR ‘*dimension*’ OR ‘aspect’ OR ‘factor’ OR ‘construct’ OR ‘individual differences’ OR ‘predict*’ OR ‘empirical’ across titles, abstracts, and keywords. This level is included to focus on retrieving quantitative research that examines curiosity by including synonyms of ‘dimensions’ in the search string. Finally, we searched for ‘scale’ OR ‘measure’ across full texts as third level keywords to narrow results to articles that include measures of curiosity. Each level of keyword(s) was connected by the Boolean AND. The finalised search terms were then used to gather records from the three selected databases (Scopus, Web of Science, and ProQuest). Searches across all three databases were also filtered to display only English language items. A total of 1120 articles were identified. A second search using the same keyword string was conducted in February 2022 and found an additional 66 articles. To capture the grey literature, highly relevant review articles were chosen and their reference lists examined to identify possibly relevant records not picked up by the database search [20,29]. A google scholar search was also conducted to source for additional grey literature. These two methods resulted in an additional eight hits, for a total of 1194 records.

### 2.4. Article Screening

To decide on the full list of articles to be used in the scoping review, two levels of screening were implemented: title and abstract screening, followed by full text screening. Prior to the screening process, EndNote 20 was used to remove 473 duplicate entries, and Rayyan was used to aid the screening process [32,33]. Two reviewers—the first author and a collaborator (both BSc Hons graduates who are proficient with article screening) assessed all records at each given stage (barring articles that had already been excluded), and disagreements were resolved via discussion and consensus. Pilot testing on 10 articles was carried out before each level of screening to ensure inter-rater reliability, and on both occasions achieved at least 90% inter-rater reliability. Likewise, inter-rater reliability of the abstract and full text screening for the complete set of articles was at least 90%. Screening of articles was conducted without excessive time constraints, which mitigated fatigue. At the abstract/title screening stage, 339 articles were excluded. A further 137 articles were omitted after screening the full texts, resulting in 245 articles that were included in the final scoping review. The breakdown of the article screening process is documented in Appendix A. 

### 2.5. Data Extraction

A data extraction form was created and preregistered as part of the scoping review protocol. The variables of interest were: article title, year of publication, author(s), sample size, sample profile, age, gender, country, curiosity measure adopted, dimensionality of adopted measure, number of dimensions and items used, response format of measure, design of study, context in which curiosity was studied, key findings, and areas for future research. The first author extracted and tabulated the data using Microsoft Excel. 

## 3. Results

### 3.1. Demographics and General Characteristics 

All 245 included articles were published between 1965 and 2022. Twenty articles (8%) were published before 2000; 34 (14%) between 2000–2009; 127 (52%) between 2010–2019; and 64 (26%) between 2020 to February 2022, when the second search was conducted. There was a strong upward trend of studies meeting the inclusion criteria, especially between 2010–2019, reflecting the rapidly increasing size of many scientific literatures [34].

With regard to the geographical origin of each record, we took the corresponding author’s institutional affiliation (or first author if no information on corresponding author was provided) if it were not explicitly stated where the study took place. Given the large number of included articles, we grouped them by continental region: North America represented 108 articles (44%); Europe contributed 52 (21%) articles; followed by Asia with 52 (22%); and Australia (11; 4%). Lastly, there were 22 articles (9%) that used mixed samples across multiple regions. In terms of specific countries, the USA contributed by far the greatest number of included articles with 103, followed by Taiwan with 11, and Australia with 10. 

The most commonly used sample was students, who were participants in 152 articles (62%), with the majority (110 articles) comprising of students who were at least at the undergraduate level. Forty-five articles (18%) used general public/community samples. Forty-six articles (18%) used more targeted samples, such as video gamers, entrepreneurs, and physicians, and 12 articles (5%) did not specify a sample profile.

### 3.2. Dimensionality of Curiosity

We classified dimensionality of curiosity into five categorizations. Firstly, articles that viewed curiosity as a unitary construct and employed a single curiosity score in their analyses were labelled as unidimensional. Secondly, articles that acknowledged the multifaceted nature of curiosity but chose to focus on one facet either by using a facet-specific measure or a subscale of a multidimensional measure were termed multidimensional (focused). Thirdly, articles with a multidimensional perspective but used an aggregate score to represent curiosity would be termed multidimensional (ignored). Fourth, articles that adopted the multidimensional view, measured various facets, and then employed analyses involving different scores for different facets were classified as multidimensional (embraced). Lastly, articles that were ambiguous in their position on dimensionality would be categorized as unclear.

Of the included articles, 34 articles (14%) treated curiosity as a unidimensional construct. For example, an article reported that a one-factor structure of curiosity (i.e., unidimensional) fitted better compared to a two-factor one [35]. Similarly, Marvin and Shohamy also reported curiosity as a single-factor construct [12]. Twenty-six articles (11%) viewed curiosity as a multidimensional construct, but investigated it in a focused manner (i.e., focusing only on one facet). For instance, articles of this nature homed in on work-related curiosity by only measuring and analysing that specific dimension, or focused only on social curiosity [36,37]. An additional 31 articles (13%), while recognising that curiosity is multidimensional, chose to use a global score, or a single-item measure. For example, Kaczmarek and colleagues utilised a multidimensional measure of curiosity but only reported an aggregate score; Halamish et al. acknowledged that curiosity is multidimensional but used a single-item curiosity measure [38,39]. One hundred and thirty-three articles (54%) acknowledged the multidimensionality of curiosity and studied multiple facets of it, such as in Kashdan et al. where a total of six dimensions were investigated and analysed separately, or in Litman’s work, where two dimensions of curiosity were examined [16,40]. The remaining 21 articles (9%) categorized as unclear—it was not explicitly stated if the researchers considered curiosity uni- or multi-dimensionally [41,42]. Figure 1 shows the breakdown of curiosity dimensions.

Across the included articles (see Figure 2), 62 articles conceptualised curiosity as having a deprivation aspect, where an individual displays curiosity to lower anxiety and frustration, such as from not knowing the answer to a puzzle. Forty-two articles conceptualised curiosity as the pleasure to discover new knowledge and gave this factor a general name called interest-based curiosity. From a more temporal point of view, 32 articles viewed state curiosity (as opposed to trait) as a dimension of curiosity. Nineteen articles conceptualised curiosity as having a dimension of stretching (seeking new information and experiences), and 18 articles conceptualised curiosity as the willingness to accept the anxiety-provoking and unpredictable nature of life (also called embracing). Other dimensions of curiosity picked up by the scoping review included the social aspect of curiosity, commonly called social curiosity, with 18 articles. Eighteen articles included specific curiosity as a dimension (solving particular and singular problems), while 17 articles viewed curiosity as having a diversive element (broad spectrum information seeking). Sixteen articles consisted of a dimension called joyous exploration, where an individual finds joy and interest in acquiring new information. Fifteen articles investigated thrill-seeking propensity as part of curiosity, while 14 articles contained the investigation of perceptual curiosity, which generally assesses individuals’ interest in sensory stimuli. Stress tolerance was also reported as a dimension of curiosity, with 13 articles. Twelve articles investigated the dimensions of exploration (tendency to pursue new information), and absorption (ability to focus full attention to tasks). Akin to social curiosity, interpersonal curiosity was reported as a dimension of curiosity in five articles, while on the flip side, intrapersonal curiosity, or curiosity about the self, was studied as part of four articles. Three articles investigated curiosity in terms of its breadth and depth (curiosity about many topics vs. sustained interest toward a single topic). The remaining records examined highly specific dimensions of curiosity (e.g., recollection of curiosity during childhood) and mostly only appeared in a low number of articles, thus will not be highlighted here. 

### 3.3. Measures of Curiosity

Of the psychometrically validated curiosity measures used, the most widely used survey was the Epistemic Curiosity Scale (ECS), which was administered in 32 articles. [16]. The next two commonly used measures were the Curiosity and Exploration Inventory-II (CEI-II; Kashdan et al., 2009) and the Epistemic Curiosity Scale, tied at 29 articles each [1,5]. The original version of the CEI-II, the CEI-I was used in 21 articles [43]. Naylor’s (1981) Melbourne Curiosity Inventory was used in 17 records, while Curiosity as a Feeling of Deprivation scale (CFD was used 15 times [44,45]. The curiosity scales (also called the State-Trait Curiosity Inventory) in the State-Trait Personality Inventory came up in 14 articles [46]. Of the two most recently developed measures, Kashdan et al.’s Five-Dimensional Curiosity Scale (5DC) was administered in 14 papers, while its updated version—the Five-Dimensional Curiosity Scale Revised (5DCR) was employed in two articles [40,47]. The remaining measures were utilised less than 10 times each (e.g., Interpersonal Curiosity Scale; Work-related Curiosity Scale), and will receive no further mention here [17,25]. With regard to non-validated measures of curiosity, a total of 23 articles used ad hoc single-item scales to measure curiosity. Figure 3 depicts the commonly used validated measures of curiosity.

Of all the curiosity measures used, 226 (92%) articles relied solely on a Likert scale response format. Twelve articles (5%) investigated curiosity using only binary questions (e.g., Curiosity about Sexual Events scale using a True/False response format; single-item curiosity measure using Yes/No) [48,49]. Four articles (2%) used a mix of Likert and binary scales, such as in Reio et al.’s work [50]. One article (0.4%) used a visual analogue scale to assess curiosity, and one article (0.4%) used a mix of Likert and visual analogue scales [51,52]. Lastly, one article (0.4%) did not specify the response format for measuring curiosity [53].

### 3.4. Context of Curiosity Research

To better understand the nature of curiosity in the literature, we were interested in whether curiosity was characterised as being context-general or studied in specific domains. Articles that were not explicit in the context, or were ambiguous, were categorised as context-general. The data extracted showed that 108 (44%) articles studied the construct of curiosity as context-general, meaning no specific context or domain was examined. Of the remaining, 69 (28%) articles focused on the workplace, such as entrepreneurship (e.g., Jeraj & Antoncic) [54] and job performance (e.g., Reio & Callahan) [3]. Twenty-nine (12%) articles revolved around education and included articles on academic performance and success (e.g., Powell and Nettebeck) [55], student development (e.g., Vracheva et al.) [56], and student learning (e.g., Wade and Kidd) [57]. 

Investigating curiosity in the context of leisure activities accounted for 26 (11%) articles. Contexts such as travelling, backpacking, and tourism (e.g., Chen & Hsu; Jani; Totsune et al.) [58,59,60], gaming (e.g., Huck et al.; Kim & Lee) [61,62], and social networking (e.g., Fang; Thomas & Vinuales) [63,64] fell under this umbrella term. Other leisure-based contexts that were less studied (compared to travelling, gaming, and social networking) included curiosity in music, virtual reality, nature, smoking, alcohol consumption, sports, visual arts, and noctcaelador (an attachment to the night sky) [26,28,65,66,67,68,69,70,71].

Five (2%) articles studied curiosity in clinical settings. For example, Denneson et al. [72] measured curiosity in their study of suicide ideation among military veterans. Other articles set in the clinical domain include therapy, myocardial infarction, and schizotypy [73,74,75]. Lastly, eight (3%) articles focused on curiosity and health, primarily on well-being [38,76,77,78,79,80], personal growth [81], and quality of life [82]. 

## 4. Discussion

The first objective of this scoping review was to map the literature on curiosity, in terms of dimensionality, how it is measured, and the context that it is studied in. The second objective was to uncover gaps that can inform future research. Before going into the theoretical implications, we first detail the relevance of curiosity research based on geographical location.

### 4.1. Geographical Location

Of the identified articles, approximately 70% of articles were conducted in Western countries (North America, followed by Europe, and Australia). Conversely, Asian countries only accounted for an estimated 22%. This suggests an imbalance of literature that skews towards Western cultures. Research has found that culture plays a role in shaping personality traits [83]. Given that curiosity has been studied as part of the web of personality traits in personality inventories such as the NEI-PI-R and the HEXACO, it is possible that curiosity differs across cultures, but this notion is obscured by researchers not specifically investigating the construct of curiosity [9,10]. Taking a finer grained approach by focusing solely on curiosity, Ye et al., using a similar sample in terms of age and gender, found that the CEI-II (developed and validated in the USA) did not conform to a two-factor structure as per the original scale, in a sample of Chinese students in Hong Kong [35]. The researchers reported that a single factor curiosity construct fitted the data better. Moreover, the Hong Kong students showed higher levels of curiosity compared to the USA sample, which could be due to living in a culture that emphasises innovation and creativity, which are common correlates of curiosity [84]. Generally speaking, there is a lack of research surrounding levels of curiosity between different cultures. Of the 22 articles that used mixed country samples, only six articles mentioned cultural impact when interpreting their findings. Moreover, of these six articles, only one involved an Asian country (Hong Kong). Accordingly, more research can be done in Eastern cultures to obtain parity and more points of comparison with their Western counterparts.

From the perspective of an individualistic-collectivistic continuum based on the cultural dimensions theory by Hofstede [85], research often points to Western countries as being more individualistic, and Asian countries as more collectivistic, and it is often the case that openness to experience (a personality factor that curiosity is nested under) is higher is Western cultures compared to Eastern cultures [86,87]. However, by using this distinction, nuances can be missed out on. For example, Singapore, located in Southeast Asia, often blends both individualistic and collectivistic principles, and is aligned with Western modernity while retaining its Asian values, such as prioritising harmony and conformity [88]. It would be insightful to investigate how curiosity works not only in an underrepresented region (Asia), but also in multicultural or blended countries. In our scoping review, there were no studies involving Singapore, and only four originating from the region of Southeast Asia (three from Malaysia, and one from Indonesia).

Secondly, again drawing upon the cultural dimensions by Hofstede [85], a difference in power distance within cultures can affect levels of curiosity. Power distance is defined as the extent (usually high vs. low) to which the less powerful members of organizations and institutions accept that power is unevenly distributed, and tends to be higher in Asian countries, while lower in English-speaking Western countries [89]. In relation to curiosity, Karandikar et al. [90] conducted a cross-cultural study of curiosity between Americans and Indians and found that the Indian sample showed lower snooping (a covert social curiosity facet) than their American counterparts. They explained that the difference in said levels of curiosity could be due to a higher power distance between individuals in India, which then led to them maintaining distance from people perceived to have higher status, ultimately resulting in individuals stifling their curiosity.

Lastly, a cultural difference in levels of curiosity can also be characterised by the high vs. low uncertainty avoidance dimension, where a high uncertainty avoidance culture (e.g., Japan) display increased stress and anxiety when faced with ambiguity, while low uncertainty avoidance cultures demonstrate the opposite (e.g., USA) [87]. Drawing links to curiosity, specifically the stress tolerance facet of curiosity, in order to elicit curiosity, one has to make a judgement that they are able to cope with the stress that accompanies uncertainty [6]. Considering research has shown that different cultures display varying degrees of uncertainty avoidance (e.g., high in East and Central European countries, low in Chinese culture countries) [85], it is plausible that people from different cultures would also possess different levels of curiosity, especially on the stress tolerance facet.

### 4.2. Dimensionality and Measurement of Curiosity

Since the dimensions of curiosity are often based on the measure of curiosity used, they will be discussed in the same section. Results from the scoping review suggest that curiosity was more commonly conceptualised as a multidimensional construct (78%) as opposed to a unidimensional one (14%). This aligned with views that there are multiple dimensions to curiosity. This was also further highlighted with the majority of articles (*n* = 133, 54%) that embraced this multidimensionality by analysing individual facets of curiosity. Curiously, research that viewed curiosity as unidimensional sometimes used the term in a more colloquial manner, rather than as a psychological construct [91]. While this may not affect the robustness of the studies, it does imply that there is much more to curiosity when viewed as a psychological construct rather than simply as a synonym of interest. There was also a small subset of articles (*n* = 31, 13%) that viewed curiosity as multidimensional, but used a global score in their analyses. While the appropriateness of using a global score is beyond the scope of this review, it is worth noting that using global scores could overinflate or underinflate the importance of curiosity due to the aggregation of scores. 

Findings from this scoping review show that the most-studied dimension of curiosity is a deprivation type of curiosity. This factor of curiosity stems from individuals trying to resolve information gaps, where they strive to seek answers to avoid unpleasant feelings of not possessing said information [16]. This then manifests as curiosity. Likewise, Kashdan et al. [6] put forward the factor of deprivation sensitivity as one of the core dimensions of curiosity. Together, these two conceptualisations of a deprivation-type curiosity made up a substantial number (*n* = 62) of the articles that conceptualised curiosity as multidimensional. 

Conversely, although the nomenclature differs, a sizable portion of included articles (*n* = 58) conceptualised curiosity as the pleasure to obtain information (joyous exploration and interest-based curiosity) [6,16]. Results from this scoping review also found that researchers studied curiosity along the dimensions of diversive (*n* = 17), specific (*n* = 18), and breadth/depth (*n* = 3). This classification conceptualised an individual’s curiosity being either more spread out across a wide range of subjects or focused on a single one. The abovementioned categorizations do not refer to the topic of interest itself, but rather the ‘how’ of the way people are curious. While investigating curiosity based on the different axes of dimensionality can provide a more holistic picture, it does not discount the importance of the context-specific nature of curiosity. 

Findings from the present scoping review suggest that the Epistemic Curiosity Scale by Litman [16] was the most common measure used, followed by the CEI-II by Kashdan et al. [1]. In the CEI-II, curiosity is split into two factors, one involving the motivation to consume new experiences (e.g., “I actively seek as much as I can in new situations”) and the other, the inclination to accept unpredictability (e.g., “I like to do things that are a little frightening”). Litman, in the Epistemic Curiosity Scale, goes one step further by incorporating a failure-avoidance factor in deprivation type curiosity (e.g., “Spend hours on a problem because I cannot rest without an answer”). Kashdan et al. in 2018, and later in 2020, sought to combine isolated strands of research into a comprehensive curiosity measure, and developed the 5DC, which was later refined into the 5DCR [6,47]. Although this has been less widely utilised than the Epistemic Curiosity Scale and the CEI-II, we believe that this is largely due to the measure having been developed only recently. The 5DC was the first to include a social curiosity factor among other existing factors. This was further expanded into overt and covert social curiosity factors in the 5DCR. Additionally, both the 5DC and its successor encompass an appraisal style factor called stress tolerance, where an individual must determine if they are able to cope with the stress that accompanies novel and unfamiliar situations, before any manifestation of curiosity. Additionally, Kashdan et al. synthesized a body of research that was largely ignored in the construct of curiosity: thrill-seeking, where individuals feel a rush of excitement whenever they have to take risks [6,47]. 

Taken together, the 5DCR represents the newest measure of curiosity that captures the full bandwidth of curiosity. This has multiple implications, such as enabling researchers to better study the correlates of curiosity using a single multidimensional scale, and also to allow for investigating curiosity profiles, or ‘types’ of curious people [6]. However, to establish that the 5DCR is truly the most robust measure of curiosity, more research should be conducted.

### 4.3. Context of Curiosity Research

Results from the present scoping review found that 108 (44%) of the included articles were context-general in nature (i.e., not investigating curiosity in specific domains). While curiosity may be trait-like and generalised across all contexts, some researchers have also put forward a more nuanced distinction, arguing that individuals exhibit both domain-general and domain-specific forms of curiosity that vary across multiple contexts [24]. From the remaining 137 (56%) articles that specified a context, roughly half were situated in a workplace context (69), followed by education (*n* = 29, 21%). Intuitively, this is unsurprising, given the importance of schooling, and thereafter joining the workforce once an individual graduates from school. Moreover, curiosity has often been implicated in both education and at the workplace. For example, curiosity aids learning by promoting engagement and improving academic performance [4], while curiosity is also positively linked with job performance and a sense of belonging to organisations [2]. 

Compared to curiosity in employment and education, much less research has been conducted on curiosity in leisure activities (*n* = 26, 19%). Based on longitudinal data, people spend a comparable amount of time between work and leisure activities [92], indicating that leisure activities represent a major domain of human activity, but have somewhat escaped the attention of curiosity researchers. Of note, there is a sharp increase in the leisure activity of video gaming, and this could present an opportune time to delve deeper into this topic, given the increased interest and curiosity to play video games ever since the COVID-19 pandemic started, where global lockdowns and quarantine conditions forced individuals to stay home [93]. On a global scale, time spent video gaming during COVID-19 has increased by 39% [94]. Six articles in the present scoping review were situated in the domain of video gaming. Given the growing number of video gamers and time spent gaming, more research can be devoted to studying curiosity in video gaming. This is perhaps made even more important, since research has pointed out that curiosity not only drives video gamers’ engagement but can also have a negative effect—video gaming addiction [95,96].

Outside of games, people also spend increasing amounts of time on social media, which is further exacerbated by a transition from offline to online activities (remote working/learning) [97,98]. Despite its many advantages such as the ease of information exchange, longer usage of social media can lead to addiction. More importantly, curiosity has been implicated as a potential antecedent to addictive social media use [99]. For example, Brailovskaia et al., found that curiosity (the desire to know what happens on Facebook) was a predictor of Facebook addiction [99]. In our scoping review, only three included articles had a social networking context. Given the ubiquitous nature of social media sites nowadays, more research should be conducted on the links between curiosity and addiction to social media. 

Based on findings from the present scoping review, only 13 (5%) of included articles were situated in the context of a clinical setting and health, and were all conducted pre-COVID-19. Along with individuals potentially turning to different leisure activities due to COVID-19, the global pandemic may have also highlighted the importance of investigating how curiosity works in clinical settings or how it is related to an individual’s well-being. Research has found that possessing high levels of curiosity increases coping efficacy to stop negative thoughts in suicidal clients [72]. Not only is curiosity useful for clients, but also for therapists—Lefevre reported that therapists who endorse higher levels of curiosity had less compassion fatigue, enabling them to better connect with patients [73]. It is well documented that COVID-19 resulted in individuals experiencing worsening mental health [100]. As such, this can serve as a call to action that more curiosity research can be done not only in the context of mental health, but also on well-being and quality of life. 

### 4.4. Limitations and Future Research

A few limitations of this scoping review should be noted. Firstly, only English language records were included, thus non-English but relevant articles could have been missed out on. The smaller number of articles originating from Asia, compared to North America, could be due to this limitation, where English may not be the first language in countries from these regions. If this holds true, findings should be interpreted with caution. This could also represent a constraint for examining how curiosity differs across cultures. However, available resources necessitated the exclusion of non-English language records. 

Additionally, in the present scoping review, methodological rigour and quality of articles were not investigated. Since the purpose of a scoping review is to map the literature, rather than assessing quality, we opted to include not only published articles, but conference papers and dissertations as well. This aligned with the main objective to obtain breadth of research being conducted on the construct of curiosity. 

As part of the data extraction form, we looked at future directions that the included articles proposed. As previously mentioned, curiosity does possess an unhealthy downside, such as addiction [95]. A total of 10 (4%) articles made recommendations to investigate the negative aspects of curiosity, such as its maladaptive effects, and individuals’ propensity to engage in risky and unhealthy behaviours. To bolster the literature, future research can seek to further examine how curiosity interacts with maladaptive behaviours. For example, Dahabiyeh et al. found that curiosity leads to individuals downplaying risks and driving intention to play online games, resulting in addiction [101]. In Lindgren et al., the researchers found that higher scores on the absorption factor of the CEI-I were associated with more alcohol-related problems, but this was not found for the exploration factor [68]. This suggests that perhaps only specific facets of curiosity are applicable to maladaptive behaviour, and can be studied as such, as compared to using global scores of curiosity. Researchers can also go one step further by investigating curiosity in the nuances among contexts of addiction. Current views suggest that gaming addiction works in a similar fashion to substance and gambling addiction [102], but the specific role of curiosity with respect to the different types of addiction is still inconclusive, and thus provides fertile ground for future research. 

The current scoping review only included records that used quantitative and self-report surveys. While self-report questionnaires are by far the most common, other forms of data can enrich the literature. For instance, qualitative data can shed light on individuals’ in-depth experience of curiosity, and their personal perspectives [103]. Studying curiosity from a neuroscience perspective can also further the field. For example, Marvin et al. reported that curiosity and impulsivity share surprising overlaps in their neural substrates. An interesting takeaway from their research involves the notion that educational and cultural consensus trend towards trying to decrease impulsivity in individuals, which could then inadvertently attenuate curiosity as well, along with its benefits [104]. Taken together, conducting curiosity research using measures other than self-report surveys can help researchers achieve a more holistic picture on how curiosity functions. 

Lastly, the 5DCR, developed by Kashdan et al. is the latest contemporary measure of curiosity, and also purported to be the most comprehensive, capturing the full construct of curiosity, and providing greater predictive power [47]. Given its recency, it is unsurprising that only two articles used the 5DCR in their research. Future research can continue to not only utilise the 5DCR, but also validate its applicability across different cultures and countries. There are currently no cross-validation studies involving the 5DCR. Considering the substantial role that culture can have in affecting curiosity, it is important to investigate the 5DCR in a non-Western culture. 

The 5DCR may also be used in conjunction with other curiosity measures that focus on facets of curiosity not measured by it. For example, Litman et al. developed a curiosity scale surrounding an individual’s curiosity about themself—the Intrapersonal Curiosity scale [17]. An introspective form of curiosity can be helpful in reaping benefits from activities such as meditation, where a major goal is the improvement of the self-concept and accepting oneself [105]. Research has shown that curiosity is positively related to individuals practicing meditation [106]. Considering that meditation focuses on the self, it is possible that an inward-looking curiosity can be efficacious in promoting personal growth and reducing self-discrepancy beyond a generic conceptualisation of curiosity (i.e., curiosity as unidimensional and context-general), even if it is through indirect means (e.g., via meditation). Whether intrapersonal curiosity can be assimilated with the 5DCR remains to be seen, but investigating this factor of curiosity is a worthwhile endeavor for future research. 

## 5. Conclusions

This scoping review provides an overview on how curiosity research has been conducted over the past six decades, highlighting how curiosity has been conceptualised, the popular measures used, and the contexts in which curiosity research has taken place. This review also offers a starting point for further research into the fascinating nature of curiosity, including investigating the downsides and many nuances of curiosity, expanding outside the scope of self-report measures, cross validating a contemporary curiosity scale, and allocating resources to intrapersonal curiosity. Enriching the curiosity literature can then potentially serve to inform ways to harness benefits to mental health and well-being, especially in the aftermath of a global pandemic. 

## Figures and Tables

**Figure 1 behavsci-12-00493-f001:**
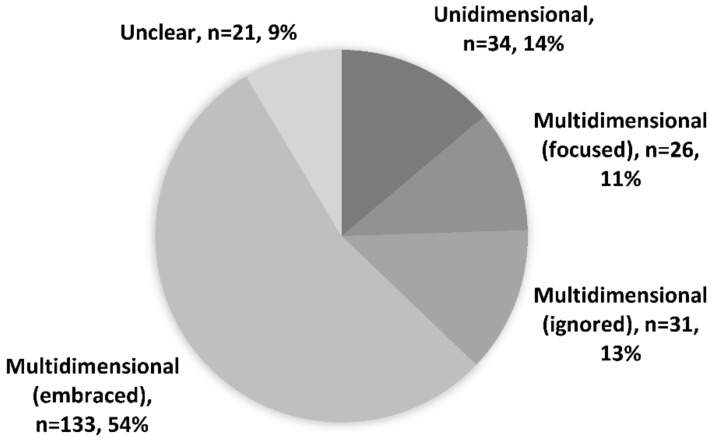
Dimensionality of curiosity.

**Figure 2 behavsci-12-00493-f002:**
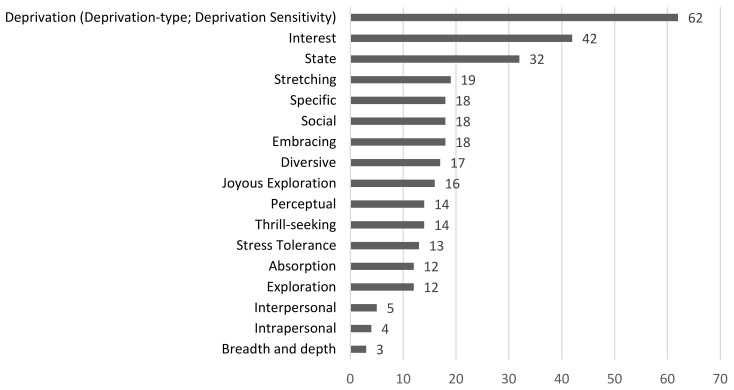
Common dimensions of curiosity examined.

**Figure 3 behavsci-12-00493-f003:**
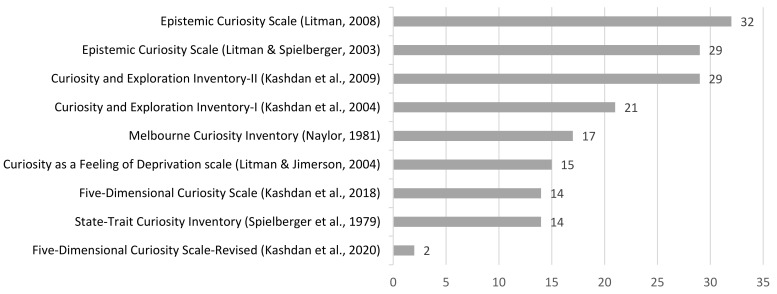
Commonly used validated curiosity measures [1,5,6,16,43,44,45,46,47].

## Data Availability

Data can be found at: https://osf.io/dj7xg/ (accessed on 8 November 2022).

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
