# Peer review of "Dimensions, Measures, and Contexts in Psychological Investigations of Curiosity: A Scoping Review"

_behavsci, 2022, doi:10.3390/bs12120493_

Round 1
Reviewer 1 Report
The subject of the researchers' interest was curiosity as a psychological variable. The authors set themselves the goal of reviewing the existing psychological literature on curiosity. They conducted a meta-analysis. The meta-analysis was based on texts from three multidisciplinary databases: Scopus, Web of Science, and the ProQuest platform. Criteria for reviewing texts/articles in databases have been defined: titles, abstracts, keywords - due to terms such as: curiosity, dimensions of curiosity, tools for researching/measuring curiosity, contexts. The inclusion criteria were therefore defined in line with the JBI recommendations, the PCC (Population, Concept, and Context) framework was used. The levels of selection of articles from the Scopus database were determined - on this basis, an "initial set" was created for further analysis. In the next step, texts from the other two databases were taken. Two competent judges were used (their training was not clearly specified) and were included in the screening study. (This raises the question of the fatigue of these two researchers.) As a result of the selection - from among the 1194 texts found - 245 articles were introduced to the last stage of the analysis.
The study refers to the principles consistent with the Preferred Reporting Items for Systematic Review and Meta-Analyzes (PRISMA).
The procedure allowed the researchers to obtain answers to the questions posed that guided the meta-analysis. Namely, the literature review indicated the most common ways of understanding curiosity, simplicity vs. content complexity of the construct, methods of research (scales, questionnaires), contexts, groups of respondents, and cultural or geographical regions (etc.)
Only texts in English were used in the analysis, although there are probably also scientific publications in other languages - which include the title, abstract and key words in English. It can be assumed that this resulted in a limitation of knowledge about the phenomenon of interest to researchers - but it was in line with the inclusion criteria adopted by them.
The researchers managed to achieve the set goal.
Author Response
Good day,
Thank you for your review of our manuscript. Kindly find below our responses to your comments.
Point 1: Two competent judges were used (their training was not clearly specified) and were included in the screening study. (This raises the question of the fatigue of these two researchers.)
Response 1: Both reviewers are BSc Hons. graduates, with knowledge in screening guidelines. Screening of articles were conducted without excessive time constraints, which mitigated fatigue. High inter-rater reliability coefficients (all >90%) across all screens conducted (pilot and complete dataset; abstract and full text) also indicated that the likelihood and impact of fatigue were minimized.
Point 2: Only texts in English were used in the analysis, although there are probably also scientific publications in other languages - which include the title, abstract and key words in English. It can be assumed that this resulted in a limitation of knowledge about the phenomenon of interest to researchers - but it was in line with the inclusion criteria adopted by them.
Response 2: This point was acknowledged and highlighted in the limitations section (page 13 of manuscript). We opted to only include texts in English due to available resources.
Reviewer 2 Report
Thanks to the authors for the interesting work!
1) The genre of the article is unusual for me, and, frankly, it is not acceptable for me personally. This genre does not fit into the tradition of empirical substantiation of arguments.
2) Nevertheless, the authors conducted a fairly correct generalization of many previous studies. And the methodology of summarizing the results of empirical research is understandable and acceptable to me.
3) The discussion of the results of generalization is, of course, partly subjective and not impartial. But this is acceptable, there is something for a simple discussion.
Author Response
Good day,
Thank you for your comments on our manuscript. Do let us know if there is anything we can improve upon, thank you.
Regards,
Yong Jie, on behalf of the authorship team